# ORDerly: Datasets and benchmarks for chemical reaction data

## Abstract

Machine learning has the potential to provide tremendous value to the life sciences by providing models that aid in the discovery of new molecules and reduce the time for new products to come to market. Chemical reactions play a significant role in these fields, but there is a lack of high-quality open-source chemical reaction datasets for training ML models. Herein, we present ORDerly, an open-source Python package for customizable and reproducible preparation of reaction data stored in accordance with the increasingly popular Open Reaction Database (ORD) schema. We use ORDerly to clean US patent data stored in ORD and generate datasets for forward prediction, retrosynthesis, as well as the first benchmark for reaction condition prediction. We train neural networks on datasets generated with ORDerly for condition prediction and show that datasets missing key cleaning steps can lead to silently overinflated performance metrics. Additionally, we train transformers for forward and retrosynthesis prediction and demonstrate how non-patent data can be used to evaluate model generalisation. By providing a customizable open-source solution for cleaning and preparing large chemical reaction data, ORDerly is poised to push forward the boundaries of machine learning applications in chemistry.

## 1 Introduction

Advancements in chemistry and material science hinge on the availability of high-quality chemical reaction data, and the advent of machine learning (ML) for science has highlighted the value that data can bring to chemistry. One important application is in the pharmaceutical industry, where figuring out *how* to make novel molecules remains a significant bottleneck, causing delays in the "make" step of the "design, make, test" cycle [1]. Making a molecule (product) includes predicting the reaction pathway (retrosynthesis) and suitable reaction conditions (e.g. solvents and reagents), and optimising for one or more outcomes such as reaction yield, selectivity, and conversion. ML is well suited to assist with these tasks, with a range of tools being developed for forward reaction prediction [2, 3, 4], retrosynthesis [5, 6, 7, 8, 9], condition prediction [10, 11, 12], yield prediction [13, 14, 15], and closed-loop optimisation [16, 17, 18].

Building reaction prediction tools requires access to large datasets for training. Historically, researchers have accessed proprietary in-house datasets or acquired the data through commercial databases such as Reaxys [19]. The advantage of commercial databases is both the scale of the datasets available (often millions of reactions) and the annotation already completed by the publishers. Yet, these datasets are not freely available to ML practitioners, stymieing advances in reaction condition prediction in both academia and industry.

Submitted to NeurIPS 2021 AI for Science Workshop.

Recently, efforts have been made to create openly-accessible databases for chemical reaction data. In particular, the Open Reaction Database (ORD) [20] is promising due to its exhaustive schema for describing chemical reaction data and breadth of data already incorporated. Yet, many of the datasets in ORD require further processing before they can be used in ML pipelines, preventing practical use. This is especially true for the largest dataset in ORD extracted from the US patent literature (the "USPTO dataset" [21]). In this work, we endeavor to close this gap.

Herein, we present ORDerly, a new framework for extracting and cleaning data from ORD, accompanied by datasets for three reaction related tasks: retrosynthesis, forward, and condition prediction. By offering an open-source and customizable solution for cleaning chemical reaction data, ORDerly aims to contribute to the development of advanced ML models in chemistry and material science.

## 2  Problem formulation

As noted by Meng *et al*. [22], reaction related tasks operate on molecules. There are numerous machine readable molecular representations [23], including molecular graphs and strings, and in this work molecules are represented as SMILES strings. Each character $m_i$ in a SMILES string represents an atom or a molecular feature (bond, branch, ring closure): $\mathcal{M} \coloneqq m_1, m_2, m_3, \ldots, m_L$, where $L$ is the total number of characters in the string. Molecules can take on one of three roles in a reaction: reactant, product, or agent. A reaction $\mathcal{R}$ transforms $N$ reactant molecules (sometimes called educts) $\{\mathcal{M}_i^{\mathcal{E}}\}_{i=1}^N$ by breaking and forming bonds to form $M$ new product molecules $\{\mathcal{M}_i^{\mathcal{P}}\}_{i=1}^M$ using $K$ agent molecules $\{\mathcal{M}_i^{A}\}_{i=1}^K$. Agents are helper molecules that enable the reaction to proceed (e.g., solvents, catalysts).

$$\mathcal{R} : \{\mathcal{M}_i^{\mathcal{E}}\}_{i=1}^N, \{\mathcal{M}_i^{A}\}_{i=1}^K \rightarrow \{\mathcal{M}_i^{P}\}_{i=1}^M, \{\mathcal{M}_i^{A}\}_{i=1}^K \tag{1}$$

Given this view of reactions, we define three different reaction related tasks in this work.

**Forward prediction** is the task of predicting the product of a reaction $\mathcal{M}^{\mathcal{P}}$ given its reactants $\{\mathcal{M}_i^{\mathcal{E}}\}_{i=1}^N$ and, potentially, agents $\{\mathcal{M}_i^{A}\}_{i=1}^K$. Probabilistically, the task is to predict the distribution $p(\mathcal{M}^{\mathcal{P}}|\{M_i^{\mathcal{E}}\}_{i=1}^N)$. While experimental evaluation in a wet lab requires expert chemists and is a time intense task, reaction outcome prediction can help as a tool to evaluate the quality of a predicted retrosynthetic route (i.e,. the probability that the reaction predicted by the single-step retrosynthesis model leads to the desired product) [24].

**Retrosynthesis** is the task of designing a sequence of $Z$ reactions $\mathcal{R}_1, \mathcal{R}_2, \mathcal{R}_3, \ldots, \mathcal{R}_Z$ that transform a set of readily available reactant molecules $\{\mathcal{M}_i^{\mathcal{E}_1}\}_{i=1}^N$ to a desired product(s) $\{\mathcal{M}_i^{\mathcal{P}_Z}\}_{i=1}^{M_Z}$. Retrosynthesis is done in the reverse direction by starting with the desired product(s) $\{\mathcal{M}_i^{\mathcal{P}_Z}\}_{i=1}^{M_Z}$ and predicting reactants $\{\mathcal{M}_i^{\mathcal{E}_Z}\}_{i=1}^{N_Z}$ that would react to form the desired product(s). The predicted

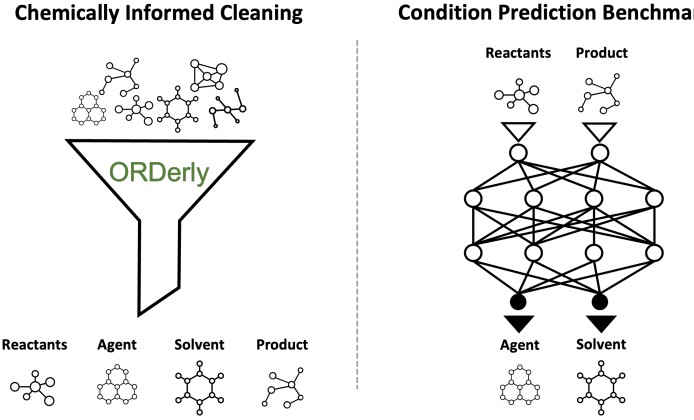

Figure 1: Overview of ORDerly.

reactants $\{\mathcal{M}_i^{\mathcal{E}_Z}\}_{i=1}^{N_Z}$ then become the products of the next reaction to be predicted $\{\mathcal{M}_i^{\mathcal{P}_{Z-1}}\}_{i=1}^{M_{Z-1}}$. This process is repeated until a readily available set of starting reactant molecules are predicted $\{\mathcal{M}_i^{\mathcal{E}_1}\}_{i=1}^N$. Therefore, the key machine learning task, often called single-step retrosynthesis, is predicting the distribution $p(\{\mathcal{M}_i^{\mathcal{E}_j}\}_{i=1}^{N_j}|\mathcal{M}^{P_j})$ or the set of reactants that could lead to a given product(s) $\{\mathcal{M}_i^{\mathcal{P}_j}\}_{i=1}^{M_j}$. Single-step retrosynthesis can be seen as the inverse of forward prediction.

**Condition prediction** is the task of predicting the distribution $p(\{\mathcal{M}_i^{\mathcal{A}}\}_{i=1}^K|\{M_i^{\mathcal{E}}\}_{i=1}^N, \mathcal{M}^{\mathcal{P}})$ (i.e., the agents for a reaction given reactants and product). In addition to agents, some models can predict continuous variables such as reaction temperature and concentrations of reactants and agents [10].

# 3 Related work

## 3.1 Chemical reaction cleaning tools

Existing tools for cleaning reaction data are primarily targeted at retrosynthesis and forward prediction tasks [25, 26, 27, 28] and have somewhat limited extensibility, given that they are built to take as inputs CSV files or the stationary XML files of the US patent (USPTO) dataset [21] instead of the outputs of continuously updated databases such as ORD [20]. Furthermore, in the original publications, there is little to no discussion of how decisions made during cleaning (e.g. restricting the number of components in a reaction or the minimum frequency of occurrence) impact the datasets being cleaned or performance of models trained on the datasets. We believe that this is in part due to data cleaning historically being viewed as a "low value" task, and therefore not adequately discussed and published on.

USPTO, being the largest open-source chemical reaction dataset, has been cleaned a number of times for different learning tasks. For example, the USPTO-50K [29, 30] and USPTO-MIT datasets [31] are commonly used for benchmarking single-step retrosynthesis and forward predictions models[1], and these benchmarks are available in aggregate benchmarking sets such as the Therapeutics Data Commons (TDC) [32]. However, the code used to process the raw data to generate the aforementioned USPTO benchmarks was not published and, there is no publicly available benchmark for reaction condition prediction extracted from these datasets.

# 4 Dataset generation

ORDerly extracts data directly from ORD [20]. Even though the data in ORD is stored in accordance with a structured schema, we found that further effort is required to transform the labeled data into ML-ready datasets. Therefore, ORDerly is centered around a data extraction script and a data cleaning script, both of which take numerous arguments that customize the operations being performed.

## 4.1 Extraction and cleaning methodology

The extraction script allows the user to choose whether reaction roles should be assigned using the labeling in ORD or using chemically-informed logic on the atom-mapped reaction string (if available). It also enables specification of data source (e.g., USPTO or non-USPTO), allowing users to train models with data from one source and test the performance with data from another source. Creating test sets from different data sources is a robust way to evaluate generalization performance.

We chose cleaning operations motivated by first-principles understanding of chemistry. Cleaning operations on the chemical reaction data include: (1) Restricting the number of reactants and product, preventing multi-step reactions being included in the datset; (2) Ensuring that all molecules can be sanitized by the cheminformatics package RDKit [33]; (3) Restricting the maximum number of unique catalysts, solvents, and reagents in a reaction based on commonly used experimental amounts; (4) Frequency filtering to remove outliers; (5) Sanity checking the yield ($0\% \leq yield \leq 100\%$), temperature, and pressure; (6) Removing duplicates, and finally; (7) Applying a random split to create

---

[1]We discuss the difference between these datasets and our dataset in Appendix A.3.2

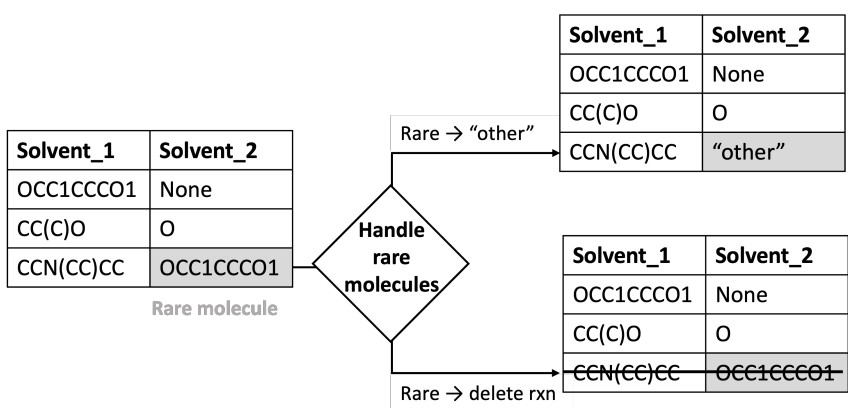

Figure 2: We present two different approaches for handling rare molecules. Rare → "other" is investigated as a strategy to avoid deleting reactions with rare molecules.

training/validation/test sets, carefully ensuring that any inputs present in the train set (i.e. reactants and products for reaction condition prediction) are not also present in the test set.

**Computational details:** All extraction/cleaning operations described in this section were performed using a 2022 Mac Studio with an Apple M1 Max chip and 32GB memory. In ORD there are roughly 1.7 million reactions from US patents (USPTO) and 91k reactions that are not from US patents. For the USPTO dataset extraction and sanitation took roughly 35 minutes, while the cleaning steps took 8 minutes.

## 4.2   Reaction role assignment

We experimented with two approaches to assigning roles to the molecules found in a reaction (e.g., whether a molecule is a reactant or an agent): trusting the labeling of molecules in ORD (referred to as "labeling") or applying chemical reaction logic to identify the role of different molecules from the reaction string (referred to as "rxn string" or "reaction string"). Our reaction logic identified reactants (molecules that contribute atoms to the product(s)) and spectator molecules (molecules that do not contribute atoms to the product(s)) based on the atom-mapping and their position in the reaction SMILES string. Solvents were identified in the list of spectator molecules by cross checking against a list of solvents compiled from prior research (see Appendix A.1.1), while all other spectator molecules are marked as agents. Catalysts were not separated into their own category since identifying catalysts can be quite subtle (especially with organocatalysis), and few reactions in the reaction string datasets contained transition metals.

## 4.3   Frequency filtering

Removing rare molecules can increase the signal to noise ratio in a dataset by removing outliers. In this work, we investigated two different strategies for filtering spectator molecules based on their frequency: deleting reactions with rare spectator molecules (rare→delete rxn) or keeping the reactions but mapping the rare molecules to an "other" category (rare→"other") (see Figure 2). We conducted experiments with both the rare→delete rxn and rare→"other" strategies for the task of condition prediction. The frequency threshold was set at 100 in line with previous research [10], though the sensitivity of dataset size to frequency threshold was still investigated (see Appendix C.2). Deleting reactions with rare molecules may create a more cohesive dataset by removing outliers, while renaming rare molecules "other" allows more reactions to be kept, offering more training data for the model.

Table 1: Number of reactions left in each dataset after cleaning. A description of each dataset can be found in section 4. Note that the actual number of reactions used for training will differ from the dataset size shown below due to train/test splits and augmentation. Non-USPTO-retro had a final dataset size of 20,830 and was cleaned in the same way as ORDerly-retro.

| Dataset name: | ORDerly-condition (labeling) | ORDerly-condition (rxn string) | ORDerly-forward | ORDerly-retro | Non-USPTO-forward |
|---|---|---|---|---|---|
| Full dataset | 1,771,032 | 1,771,032 | 1,771,032 | 1,771,032 | 91,067 |
| Too many reactants | 1,470,060 | 1,631,394 | 1,743,585 | 1,631,394 | 43,845 |
| Too many products | 1,329,399 | 1,593,196 | 1,740,655 | 1,593,196 | 40,770 |
| Too many solvents | 1,222,381 | 1,388,312 | 1,689,445 | NA | 36,522 |
| Too many agents | 1,202,790 | 1,279,833 | 1,550,800 | NA | 31,187 |
| No reactants/products | 1,202,758 | 1,262,333 | 1,533,680 | 1,567,697 | 31,095 |
| No solvents | 870,888 | 950,189 | NA | NA | NA |
| No agents | 135,139 | 690,234 | NA | NA | NA |
| Inconsistent yields | 126,948 | 658,071 | NA | NA | NA |
| Dropping duplicates | 76,634 | 392,996 | 919,231 | 941,566 | 28,496 |
| Frequency filtering | 75,033 | 356,906 | NA | NA | NA |

## 4.4 Dataset composition

Datasets generated with ORDerly have the following column groups: Reaction SMILES (string), is_mapped (bool), Reactants & products (SMILES strings), Solvents and agents (rxn string data), or solvents, catalysts, and reagents (labeling data) (SMILES strings), Temperature, reaction time, yield (floats), Procedure details (string), Grant date (datetime), date of experiment (datetime), file name (string).

Three new benchmarks were created from the USPTO dataset: `ORDerly-forward` for forward prediction, `ORDerly-retro` for retrosynthesis prediction, and `ORDerly-condition` for reaction condition prediction. Several additional datasets were created, including datasets from non-USPTO data in ORD and datasets to investigate data labeling and frequency filtering. An overview of the datasets and benchmarks showing how each cleaning step impacted the dataset size can be found in Table 1. The datasets are freely available and can be downloaded immediately from FigShare or regenerated using the code in the ORDerly Github repository.

## 5 Results and discussion

Experimental evaluation of the `ORDerly-forward` and `ORDerly-retro` benchmarks was performed using the Molecular Transformer architecture built by Schwaller *et al.* [2]. To switch from forward prediction to retrosynthesis prediction no changes to the transformer architecture were necessary, only the data was changed. The `ORDerly-condition` benchmark was evaluated together with the impact of different approaches to reaction role assignment and frequency filtering using the neural network architecture built by Gao *et al.*[10] with only minor modifications.

### 5.1 Forward and retrosynthesis prediction with transformers

Transformers were applied to two tasks: forward prediction (predicting products given reactants, solvents, and agents) and retrosynthesis (predicting reactants given a product). For the task of forward reaction prediction two different modes were tested: mixing the reactants, solvents, and agents, or weakly separating the reactants from the solvents and agents with a ">" token. Forward prediction with mixed inputs is a more difficult task, since it is less obvious which atoms (characters) will appear in the product.

For both forward and retrosynthesis prediction the order of the molecules was randomized, and the dataset was augmented by replacing each SMILES string in the reaction with a random equivalent

Table 2: Test performance with Molecular Transformer on forward prediction and retrosynthesis (%). The first column shows the percentage of invalid SMILES strings produced by the transformer (lower is better), while the second and third column show the top-1 accuracy with and without consideration of stereochemistry (SC), respectively (higher is better).

| Test sets: | Random split from USPTO | | | Non USPTO | | |
|---|---|---|---|---|---|---|
| Tasks | Invalid SMILES | Accuracy (with SC) | Accuracy (w/o SC) | Invalid SMILES | Accuracy (with SC) | Accuracy (w/o SC) |
| Forward (separated) | 0.46 | 82.18 | 84.31 | 0.31 | 82.61 | 83.62 |
| Forward (mixed) | 0.47 | 80.79 | 82.86 | 0.31 | 82.61 | 83.62 |
| Retrosynthesis | 0.25 | 49.96 | 50.99 | 0.09 | 42.28 | 42.47 |

SMILES string (thus doubling the dataset size), before finally being tokenized [2]. Performance metrics are reported in Table 2, showing that across all tasks only a small percentage of the generated SMILES strings are invalid.

On the forward prediction tasks, the accuracies achieved are similar (albeit slightly lower) to the accuracies reported by by Schwaller et al. [2] (88-90% top-1 accuracy when trained on the USPTO_MIT [31] dataset), though the accuracies are not directly comparable since different subsets of USPTO were used. As expected, the performance with separated agents is higher than mixed, since it is an easier task, and it is encouraging to see that the models get stereochemical information correct most of the time. Accuracy with the retrosynthesis model on the held out test set was roughly 50%, which is similar previous work on retrosynthesis [34]. It is interesting that prediction accuracy on the non-USPTO data was similar on the forward prediction tasks, but markedly worse on the retrosynthesis task.

**Computational details:** The transformer models were trained for around 35 hours (roughly 600 epochs) on a T4 cloud GPU instance provided by lightning.ai. Evaluation was done with the final model checkpoint.

## 5.2 Reaction condition prediction with neural networks

The reaction condition prediction model used in this work predicts five categorical variables: two solvents and three agents. These five molecules form a set (order invariant), though the loss function in the model used to predict the molecules considers them sequentially (with order) since this was found to work better in practice [10]. The metric used to evaluate the accuracy of the model should be order invariant, since the problem is order invariant, and for this reason the accuracy metrics used are top-1 (see appendix B) and top-3 (see Table 3) exact match combination accuracy for each type of component (i.e., solvent, agent). Beam search was used to identify the top-3 highest probability sets of reaction conditions. The top-3 accuracy was compared to the baseline predictive accuracy of simply predicting on the test set the most common molecules found in the train set.

Additionally, we define a metric inspired by Maser *et al.* [12] called the average improvement over baseline (AIB%):

$$AIB\% = \frac{A_m - A_b}{1 - A_b} * 100 \tag{2}$$

where $A_m$ is the exact match combination accuracy of the model and $A_b$ is the exact match combination accuracy of choosing the top 3 most common values of a component in the respective train set.

Table 3 shows the predictive performance on the test set using four different flavours of the `ORDerly-condition` benchmark. All models show an improvement over the frequency informed baseline.

Table 3: Top-3 metrics on condition prediction with the model architecture of Gao et al. [10]: frequency informed guess accuracy // model prediction accuracy // AIB%.

| Datasets: | labeling rare→"other" | labeling rare→delete rxn | reaction string rare→"other" | reaction string rare→delete rxn |
|---|---|---|---|---|
| Solvents | 47 // 58 // 21% | 50 // 61 // 22% | 23 // 42 // 26% | 24 // 45 // 28% |
| Agents | 54 // 70 // 35% | 58 // 72 // 32% | 19 // 39 // 25% | 21 // 42 // 27% |
| Solvents & Agents | 31 // 44 // 19% | 33 // 47 // 21% | 4 // 21 // 18% | 5 // 24 // 21% |

The performance of the labeling datasets at first appears to be better than those that use our custom logic to extract reaction components from the reaction string. However, as shown in Figure 5, many of the reactions in datasets where we trust the labeling in ORD have more than three reactants, while most reactions in organic chemistry only have two reactants. Upon manual inspection, we found that many agents were mislabeled as reactants and, therefore, the prediction problem was made significantly easier. This insight is confirmed in Table 4; there are fewer unique solvents and agents and a higher density of null components when using the ORD labeling instead of the reaction string. This discrepancy demonstrates that naive creation of datasets based on ORD can lead to inflated performance metrics. In dealing with rare spectator molecules to avoid sparse OHE (see Table 1) we found that rare → delete rxn strategy performed better in practice. Therefore the ORDerly-condition benchmark uses the reaction string to assign reaction roles with the rare → delete rxn strategy.

For the datasets that extract the components from the reaction string, overall top-3 accuracy is less than 25% across solvents and agents. While not directly comparable, our overall accuracy is lower than what Gao *et al.* [10] achieved with 50.1% top-3 accuracy across catalysts, solvents and agents. However, Gao *et al.* trained on approximately ten million reactions, while we train on less than four percent of that (∼350k). As shown in Figure 3, we see consistent increases in AIB (%) with the number of data points for the dataset which uses reaction strings and deletes rare reactions, and this scaling performance indicates that as ORD grows, better performance could be achieved, even with potentially fewer data points than used in the paper by Gao *et al.*

**Computational details:** These models were trained on an A10G cloud GPU instance provided by lightning.ai for 100 epochs to minimize cross entropy loss for each reaction component. The best model by validation loss was chosen for evaluation.

## 6 Technical limitations

### 6.1 Component labeling

Identifying the role of molecules in a reaction provides crucial context to machine learning models, and this identification could be improved with better atom-mapping [35]. However, an atom-mapping algorithm was not integrated into ORDerly to keep ORDerly lightweight. Even with perfect atom-mapping reaction role identification [29] can be challenging since the role of a molecule depends

Table 4: Diversity in the datasets. Frequency filtering was applied for the solvents and agents to create a more dense one-hot encoding. Columns: Number of unique molecules with a frequency above the threshold; number of unique molecules with a frequency below the threshold; percentage of the dataset that is None.

| | labeling | | | reaction string | | |
|---|---|---|---|---|---|---|
| Reactants | 40,020 | 0 | 25.7% | 317,184 | 0 | 18.4% |
| Products | 38,816 | 0 | 0.0% | 382,850 | 0 | 0.0% |
| Solvents | 29 | 204 | 40.0% | 85 | 313 | 28.0% |
| Agents | 48 | 447 | 56.2% | 255 | 11,945 | 37.0% |

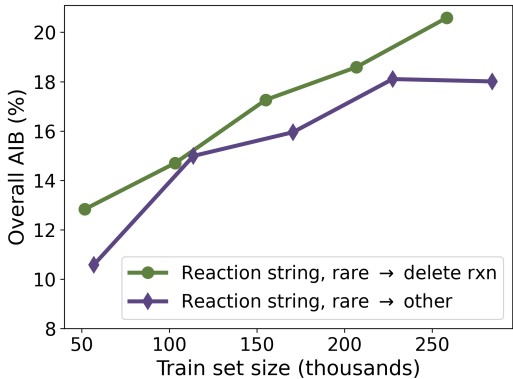

Figure 3: Scaling behaviour of different datasets with respect to overall top-3 AIB (%) for all solvents and agents (third row from Table 3.)

on the context. Reaction roles can more easily be identified when only considering one reaction class at [12], since this allows the mechanistic details of the reaction class [36, 37, 38] to be considered. Handling large and diverse datasets inevitably requires generalizations that may result in contradictions upon a more fine-grained inspection.

## 6.2 Order invariance

Although order of addition may play a role in wet lab chemistry, reaction prediction tasks are often cast as order invariant, where the goal is to predict a set of molecules. However, both of the architectures used for experimental validation of the ORDerly datasets are not agnostic to the ordering of the targets, since the neural networks used predict one molecule at a time in the OHE, and the transformers used predict one token at a time. Incorporating order invariance (and canonicalization) of the molecules into the loss calculation during training may allow for better generalisability of the predictive models, and is an exciting area for further study. It is worth noting that the evaluation metrics used throughout are order invariant.

## 7 Conclusions

In this work, we presented ORDerly, an open-source framework for preparing chemical reaction data stored in the Open Reaction Database (ORD) for machine learning applications. ORDerly was used to generate benchmark datasets for forward prediction (`ORDerly-forward`), retrosynthesis (`ORDerly-retro`), and condition prediction (`ORDerly-condition`) based on US patent data. Transformer models were trained on the forward prediction and retrosynthesis datasets, and they were found to only generate invalid SMILES strings very infrequently, while also achieving similar test accuracy to that found in the literature on a held-out set of US patents. To further investigate model generalisation ORDerly was used to generate test sets from all non-patent data from ORD, and for the forward prediction task the accuracy was comparable, while the accuracy was slightly lower for the retrosynthesis task. The condition prediction task was used to investigate different strategies for assigning reaction roles and frequency filtering of the spectator molecules. When building datasets for condition prediction using the labeling in ORD, we found contamination of the inputs (reactants) with the outputs (agents), resulting in a problem that was unrealistically easy. We therefore chose to use chemically informed logic to better assign reaction roles for the `ORDerly-condition` benchmark.

All benchmarks and datasets experimented with in this work, as well as the code used to generate them, are freely available online, and we hope the benchmarks will make reaction prediction tasks more accessible. ORDerly presents a fully open-source pipeline to go from raw ORD data to a fully trained condition prediction model, allowing for an avenue to leverage the growing contributions to open source chemistry.

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

## Appendix (ORDerly: Datasets and benchmarks for chemical reaction data)

## A  Dataset extraction and cleaning

In the main paper, we describe the "labeling" and "reaction string" datasets; in the code this is denoted by `trust_labeling=True`, and `trust_labeling=False`, respectively. We also presented two different strategies for dealing with rare molecules, either "rare→"other" and "rare→delete rxn", these are denoted in the code as `map_rare_molecules_to_other=True`, and `map_rare_molecules_to_other=False`, respectively. There are a number of other tuneable parameters in the scripts, and below we explain how default values were chosen for each of these.

### A.1  Extraction script

There are three fields in the Open Reaction Database schema to extract molecules from: the input, the outcome and the reaction string. Molecules in the reaction string are represented as SMILES, while molecules in the input and outcome field can be represented with a number of different representations, including SMILES, InChI, and plain text English names. When extracting molecules from the input or outcome field, the preferred representation was SMILES. However, how should the situation where only an English name exists be dealt with? It is tempting to check whether the representation is interpretable by RDKit (potentially implying that the molecular representation was mislabeled as a name rather than SMILES), however, this can lead to unexpected behaviour. As an example, the string, "1400C", was encountered as the name for a molecule, should this be interpreted as a graphene structure, a typo for carbon-14, a typo for 1400°C, or simply carbon? Another situation which was encountered was BOC; this *is* a resolvable smiles string, representing boron oxygen and carbon bonded together, however, in context, it was actually referring to a BOC group (tert-butyloxycarbonyl protecting group). Another example of unintended behaviour is the case of II, which could mean diiodine, but also mark the second step/item when counting. Therefore, when the user decides not to trust the labeling of the molecules, molecules only represented with a plain text name were ignored, to avoid ambiguity.

The extraction script generates the relevant data from each ORD file, and allows for the following customization Note that we only mention the arguments that materially affect the science/logic of how cleaning is done.

- `trust_labeling`: If True, maintain the labeling of the data in ORD. If False: chemical logic (described extensively in the paper) is applied to the reaction string to determine the reaction role of molecules.

- `solvents_path`: If the user does not trust the labeling, all agent molecules are cross-checked against a set of industrially relevant solvents, and any matches are re-labeled as solvents. See section A.1.1 for how this set of solvents was constructed.

- `name_contains_substring`: Only extract filenames from ORD that includes this string. If left empty will not search for anything, and if set to None it will extract data from all ORD files in the designated folder. For example, setting `name_contains_substring="uspto"` will grab all files that have "uspto" in the file name (i.e. the USPTO data).

- `inverse_substring`: The inverse of `name_contains_substring`, e.g. setting `inverse_substring="uspto"` will grab everything *except* the USPTO data.

### A.1.1  Building a set of solvents

The solvents set can be found in `orderly/data/solvents.csv` in the ORDerly GitHub repository. The set was created from the intersection of solvents from the following three sources:

- Machine learning and molecular descriptors enable rational solvent selection in asymmetric catalysis (458 solvents) [39]

- ACS Green Chemistry: Solvent Selection Tool (272 solvents) [40]

- Summit GitHub Repository (115 solvents) [41]

After the data from these three sources were concatenated into a new CSV files, the solvents were filtered by: making all solvent names lower case, stripping spaces, and then removing duplicate names. (Before removing duplicates: 458+272+115=845 solvents. After removing duplicates: 615 solvents.) Then Pura was run to resolve the solvent name where no SMILES string was available. Each solvent (with no SMILES string) was represented with up to four different names: three English solvent names (synonym names) and one CAS number. Pura was used with `services=[PubChem(autocomplete=True), Opsin(), CIR()` and `agreement=2` on each English name, and `services=[CAS()` with `agreement=1` on the CAS number. This yielded up to four different SMILES strings for each solvent. SMILES strings with full agreement for a solvent were trusted, and any rows with disagreement between the SMILES strings ($\approx 40$ solvents) were resolved by hand. The final solvents set is a CSV file with seven columns: up to three different English solvent names (synonyms), a CAS number, a chemical formula, SMILES, and finally the source.

An obvious drawback of identifying solvents by crosschecking against a curated set is that the set naturally will be incomplete; there are unfathomably many different organic molecules, and it is unclear how many of these could act as solvents. However, not distinguishing between solvents and agents may make the learning task more difficult for machine learning models, and using the labeling that already exists in ORD was routinely found to be inaccurate. In practice, the vast majority of solvents used in industry and academia are inspired by what has previously proven successful, and thus the solvents set curated for this work is likely going to capture a majority of solvents. Another difficulty is that the role of solvent molecules may depend on the context (e.g. polar protic solvents may contribute protons to the product, in which case the role of the molecule becomes murky (i.e. is it a reactant since it contributed atoms to the product, is it a solvent since it dissolved the (other) reactants, or is it a reagent since it acts like an acid?).

## A.2    Cleaning script

- `remove_reactions_with_no_reactants [bool]`: Self-explanatory

- `remove_reactions_with_no_products [bool]`: Self-explanatory

- `consistent_yield [bool]`: If True, removes reactions that have yields that do not make sense, e.g. if any individual yields, or the sum of yields, is outside of [0%; 100%] (reactions with no yields are kept).

- `num_reactant, num_product, num_solv, num_agent, num_cat, num_reag [int]`: The maximum number of components allowed of the specified type in a reaction. E.g. if `num_solv=2` any reactions with 3 or more solvents will be dropped from the DataFrame. See section C for how the default values were chosen.

- `min_frequency_of_occurrence [int]`: The frequency of molecules across all columns of the same type (e.g. solvents) are counted, and any reactions containing molecules below the frequency cutoff are dealt with in accordance with `map_rare_molecules_to_other`. See section C for how the default values were chosen.

- `map_rare_molecules_to_other [bool]`: If False, any reactions containing molecules that fall below the threshold will be deleted. If True, the rare molecules will be mapped to a string "other", allowing us to keep the reactions in the dataset. This behaviour can be shut off simply by setting `min_frequency_of_occurrence=0`.

- `set_unresolved_names_to_none_if_mapped_rxn_str_exists_else_del_rxn,` `remove_rxn_with_unresolved_names, set_unresolved_names_to_none [bool]`: These three bools control the handling of unresolvable names (i.e. names that are unresolvable by RDKit, and do not exist in our manually curated name resolution dictionary, and at most one of them can be True (if all are set to False, unresolvable names are kept in the dataset.) While the second and third bool are self-explanatory, this is the logic applied

Table 5: Comparison between different datasets for retrosynthesis and forward prediction.

| Dataset | Size | Split | Reference |
|---|---|---|---|
| USPTO-50K | 50 016 | Random | [29] |
| USPTO-MIT | 479 035 | Random | [31] |
| USPTO-full | 997 415[2] | Random | [32] |
| ORDerly-retro | 941 566 | Random | This work |
| ORDerly-forward | 919 231 | Random | This work |

if the first bool is True: if a reaction contains a mapped reaction, the reaction is seen as quite trustworthy, and therefore the unresolvable names can safely be set to None, while the remaining data associated with that reaction is maintained; if a reaction does not have an associated mapped reaction, the presence of an unresolveable name is a red flag casting doubt on the veracity of that reaction, and thus the whole reaction (a row in the DataFrame) is removed).

### A.3 Further justification for cleaning thresholds

#### A.3.1 Condition prediction benchmark

- **Reactant filtering:** Reactions with more than two reactants were filtered out, since they are likely to be multi-step reactions or complex one-pot reactions (tri-molecular mechanisms are exceedingly rare in chemistry).

- **Product filtering:** Reactions with multiple products were also filtered out since nearly all reactions in USPTO only report one product (see Figure A5); predicting reaction side products and impurities remains an active area of research [42], and thus fell beyond the scope of ORDerly.

- **Solvent and agent filtering:** Thresholds for the number of spectator molecules was set at two solvents and three agents to have the same number of categorical variables as in the model of Gao et al. [10].

- **No conditions filtering:** Reactions will not work without a solvent, and will usually require an agent. There are exceptions to this (e.g. the Diels-Alder reaction), however, the number of reactions with an erroneous recording of no agents is likely going to outnumber the amount of genuine exceptions. These filtering steps imply that a model trained on the ORDerly-condition benchmark may be ill-equipped to deal with reaction impurities or make predictions for reactions with no agents. It is worth noting that these drawbacks may be relatively inconsequential, since a skilled chemist is unlikely to query a model to predict agents for a class of reaction that requires no agents.

- **Not predicting temperature:** Only 192k out of 323k reactions in the ORDerly-condition training set contain a temperature, of which over half report 25C. Filtering away reactions without a temperature would leave a much smaller dataset, and we do not believe that it is reasonable to assume that reactions without a reported temperature were performed at room temperature.

#### A.3.2 Forward prediction and single-step retrosynthesis benchmarks

The ORDerly-retro dataset is compared to other standard forward prediction and retrosynthesis datasets in Table 5. USPTO-50K was created by Schneider *et al.* for testing reaction role assignment [29]. They used NameRxn to assign reaction classes to all the reactions in the dataset. Liu *et al.* [30] then used the USPTO-50K for benchmarking their retrosynthesis model, however, they did not use the reaction classes to create a split, and instead opted for a random split. Coley *et al.* [43] is often cited for their train/test split of USPTO-50K. USPTO-MIT is a larger set that was introduced by Jin *et al.* [31].

- **Forward prediction:** A small number of reactions in USPTO reported two products, and for the forward prediction dataset we allowed up to two products and three reactants, solvents, and agents.

- **Retrosynthesis prediction:** In retrosynthesis prediction the goal is to predict reactants that can be used to form a desired product. To ensure that the difficulty of the task was reasonable, we limit reactions to having one product and two reactants, such that the models only have to learn how to break one molecule into two, and not consider e.g. multi-producut or multi-step reactions. Only product and reactant molecules were used in the retrosynthesis dataset, so there were no restrictions in the number of solvents and agents.

## B  Further experimental details

### B.1  Condition prediction with neural networks

The code from *Gao et al.* [10] was used for training condition prediction models. The hyperparameters in Table 6 were used, which reflect those used in the original paper. Training on an A10G required 30 minutes or less for a full training run.

Table 6: Hyperparameters used for training condition prediction models

| | |
|---|---|
| batch size | 512 |
| learning rate | 0.01 |
| hidden size 1 | 1024 |
| hidden size 2 | 100 |
| dropout | 0.2 |
| fingerprint size | 2048 |

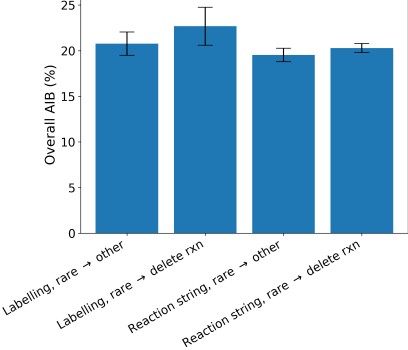

Figure 4: AIB (%) on the test sets for each training dataset. Error bars are with respect to the random seed in splitting the training and validation data (test data stayed the same).

### B.2  Forward prediction and retrosynthesis prediction with transformers

Most of the hyperparameters used in the Molecular Transformer architecture (see Table 7) were the defaults suggested by Schwaller *et al.* [2] (GitHub). The transformer models were trained for around 35 hours (approximately 600 epochs).

Table 7: Hyperparameters for Molecular Transformer.

Training

|  |  |
|---|---|
| seed | 42 |
| param_init | 0 |
| param_init_glorot | |
| max_generator_batches | 32 |
| batch_size | 4096 |
| batch_type | tokens |
| normalization | tokens |
| max_grad_norm | 0 |
| accum_count | 4 |
| optim | adam |
| adam_beta1 | 0.9 |
| adam_beta2 | 0.998 |
| decay_method | noam |
| warmup_steps | 8000 |
| learning_rate | 2 |
| label_smoothing | 0.0 |
| layers | 4 |
| rnn_size | 256 |
| word_vec_size | 256 |
| encoder_type | transformer |
| decoder_type | transformer |
| dropout | 0.1 |
| position_encoding | |
| share_embeddings | |
| global_attention | general |
| global_attention_function | softmax |
| self_attn_type | scaled-dot |
| heads | 8 |
| transformer_ff | 2048 |

Inference

|  |  |
|---|---|
| batch_size | 512 |
| replace_unk | |
| max_length | 200 |
| beam_size | 5 |

## C  ORDerly benchmark statistics

### C.1  Number of components

Figure 5 shows the distribution in the number of components of the unfiltered datasets, allowing us to compare the reaction string datasets to the labeling datasets. The distributions look quite similar for products and solvents. However, the distributions are different for reactants and agents/catalysts, which can be explained by reagents routinely being labelled as reactants in ORD.

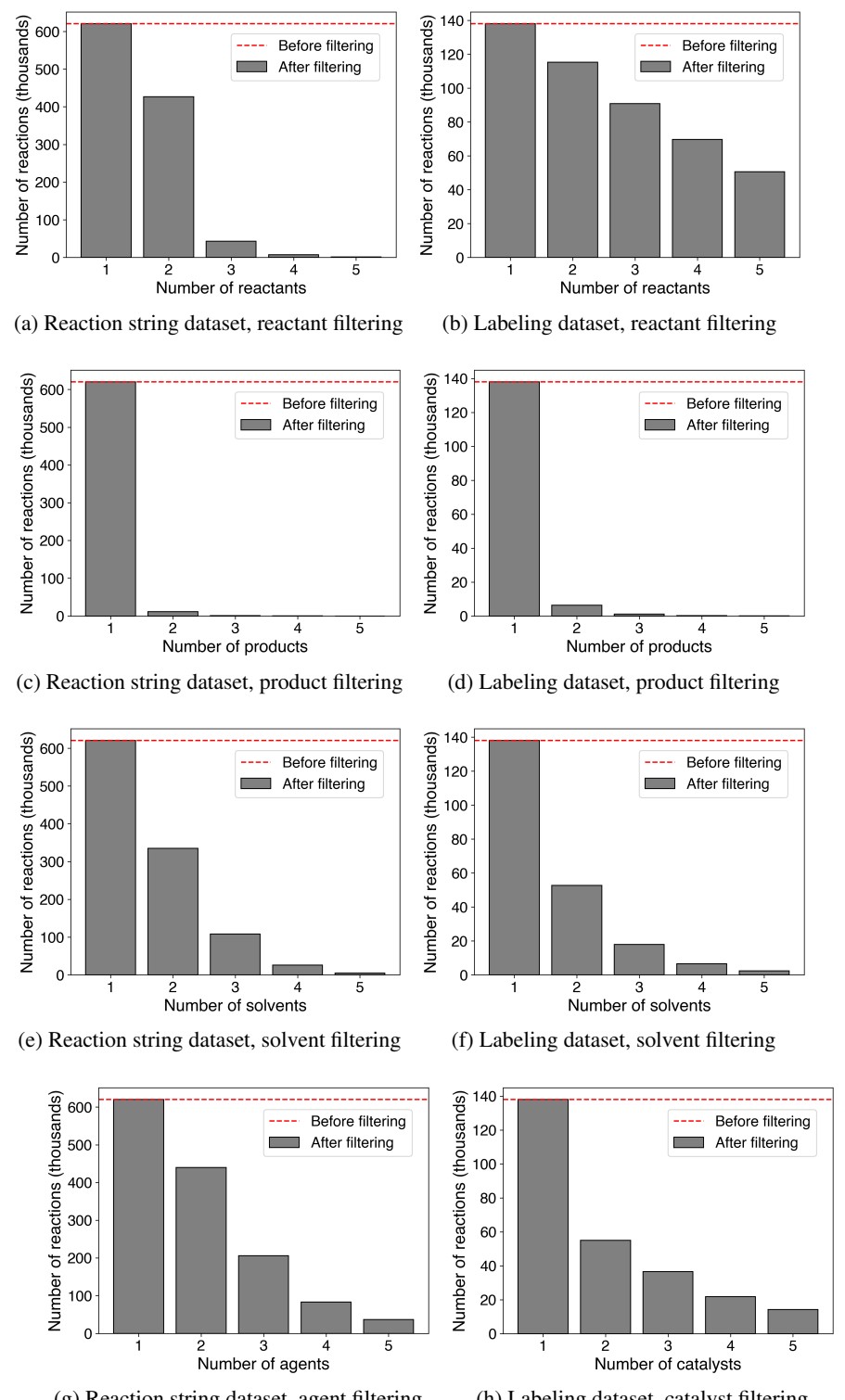

(a) Reaction string dataset, reactant filtering

(b) Labeling dataset, reactant filtering

(c) Reaction string dataset, product filtering

(d) Labeling dataset, product filtering

(e) Reaction string dataset, solvent filtering

(f) Labeling dataset, solvent filtering

(g) Reaction string dataset, agent filtering

(h) Labeling dataset, catalyst filtering

Figure 5: Distribution of the number of components between the reaction string and labeling datasets. There are no reagents in the labeling dataset, so after filtering excess catalysts were re-labelled as reagents.

 **C.2    Minimum frequency of occurrence**

551  Figure 6 shows how many reactions would be left in the reaction string and labeling datasets as
552  a function of the minimum frequency of occurrence. The minimum frequency of occurrence is
553  the threshold applied to the spectator molecules (solvents, agents, reagents, agents, catalysts) to be
554  considered rare, and any reactions containing a rare molecule will be deleted if (rare→delete rxn).

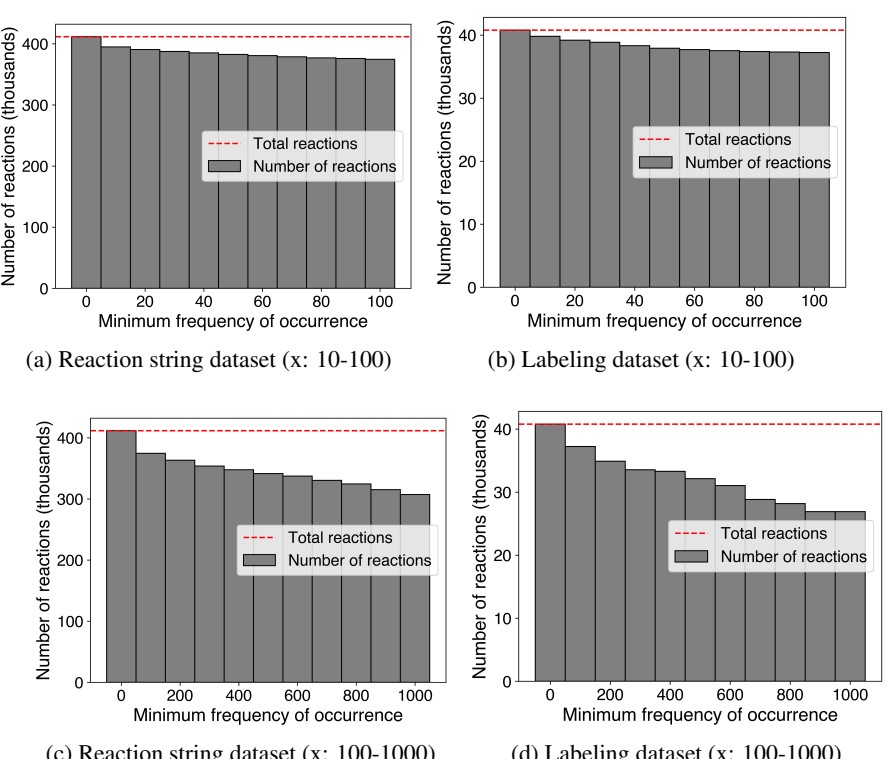

(a) Reaction string dataset (x: 10-100)          (b) Labeling dataset (x: 10-100)

(c) Reaction string dataset (x: 100-1000)        (d) Labeling dataset (x: 100-1000)

Figure 6: Impact on dataset size by changing the minimum frequency of occurrence.

555  **C.3    Molecule popularity**

556  Figure7 shows the distribution of occurrence of the top 100 most popular molecules across the
557  different categories of molecules for the labeling and rxn string datasets. Across categories, the
558  reaction string dataset is more diverse and not as heavily dominated by the most popular component.
559  It is also interesting that the most popular molecules between the datasets are not the same, despite
560  being based on the same raw data.

561  # D    Example reaction instances and predictions

562  In this section, we give examples of reactions that are in both the trust labeling and reaction string
563  datasets (Table 3) to demonstrate the differences between the different cleaning methodologies.

Reaction string input:
CC(C)(C)C(=O)c1ccccc1[N+](=O)[O-]>CCO.Cl.[Pd]>CC(C)(C)C(=O)c1ccccc1N

| | Reaction string dataset (This work) | Trust labelling dataset |
|---|---|---|
| Reactants |  |  |
| Products |  |  |
| Ground truth solvents |  |  HCl |
| Ground truth agents | Pd, HCl | Pd |
| Predicted solvents |  ✔ |  ✘ |
| Predicted agents | Pd ✔ | Pd ✔ |

**Reaction type**: Reduction (of a nitro group)

**Comment**: Trust labelling predicted the wrong solvent, which, however, can still serve as a solvent due to similar properties (polar and protic). It must be noted that the agent prediction was incomplete - no strategy predicted HCl as an agent which is crucial to enable the reaction and serve as a proton source.

✔ Correct prediction    ✘ Incorrect prediction    ✔ Partially correct prediction

565

Reaction string input:
O=C(O)c1c(F)ccc([N+](=O)[O])c1F.[H][H]>CO.[C].[Pd]>Nc1ccc(F)c(C(=O)O)c1F

| | Reaction string dataset (This work) | Trust labelling dataset |
|---|---|---|
| Reactants |  |  H-H |
| Products |  |  |
| Ground truth solvents | ——OH | ——OH |
| Ground truth agents | C, Pd, H-H | C, Pd |
| Predicted solvents | ✗ (ethanol) | ✗ (ethanol) |
| Predicted agents | Pd ✓ | Pd ✓ |

**Reaction type**: Hydrogenation (of a nitro group)
**Comment**: Hydrogen gas can be either categorized as reactant or as agent – here the approaches vary depending on the dataset. In both cases ethanol is predicted which, however, can still serve as a solvent due to similar properties (polar and protic). It must be noted that the agent prediction was incomplete - no strategy predicted H2 as an agent which is crucial to enable the reaction and serve as a hydrogen source.

✓ Correct prediction   ✗ Incorrect prediction   ✓ Partially correct prediction

Reaction string input:

CC(=O)O.Cc1nc(N2CCN(S(=O)(=O)c3ccc(OC(F)(F)F)cc3)[C@@H](C(=O)OCc3ccccc3)C2)sc1C(=O)OC(C)(C)C>CO.[C].[Pd]>Cc1nc(N2CCN(S(=O)(=O)c3ccc(OC(F)(F)F)cc3)[C@@H](C(=O)O)C2)sc1C(=O)OC(C)(C)C

|  | Reaction string dataset (This work) | Trust labelling dataset |
|---|---|---|
| Reactants |  |  |
| Products |  |  |
| Ground truth solvents |  |  |
| Ground truth agents | C, Pd | C, Pd |
| Predicted solvents |  ✗ |  ✗ |
| Predicted agents | Pd ✓ | Pd ✓ |

**Reaction type**: Acidic ester cleveage

**Comment**: We observed differences in categorizing the acetic acid as either reactant or solvent. Chemically, it should be considered a reactant or agent. In both cases ethanol is predicted which can still serve as a solvent due to similar properties (polar and protic). In the case that acetic acid is not passed as reactant the model should also predict it as agent.

✓ Correct prediction    ✗ Incorrect prediction    ✓ Partially correct prediction

Reaction string input:
CCOC(=O)c1ccc(-n2cc(C#N)c(-c3ccccc3OCc3ccccc3)c2)cc1OCOC.CCOC(C)=O
>CO.[C].[Pd]>CCOC(=O)c1ccc(-n2cc(C#N)c(-c3ccccc3O)c2)cc1OCOC

| | Reaction string dataset (This work) | Trust labelling dataset |
|---|---|---|
| Reactants |  |  |
| Products |  |  |
| Ground truth solvents |  |  |
| Ground truth agents | C, Pd | C, Pd |
| Predicted solvents |  ✗ |  ✓ |
| Predicted agents | Pd ✓ | Pd ✓ |

**Reaction type**: Ether cleveage (cleaving an Obn protection group)
**Comment**: Acetyl acetate is categorized either as solvent or reactant. Here both roles makes sense chemically. For the prediction using reaction string dataset it must be noted that while EtOH is predicted, the ground truth solvent is ethylacetate. However, under acidic conditions acetyl acetate can fall apart into acidic acid and EtOH.

✓ Correct prediction     ✗ Incorrect prediction     ✓ Partially correct prediction

Reaction string input:
CCO.Cc1c(-c2ccccc2)c(CC(=S)N(C)C)c2oc(C3CC3)nc2c1C#N>C1CCOC1.[Ni]>
Cc1c(-c2ccccc2)c(CCN(C)C)c2oc(C3CC3)nc2c1C#N

| | Reaction string dataset (This work) | Trust labelling dataset |
|---|---|---|
| Reactants |  |  |
| Products |  |  |
| Ground truth solvents |  |  |
| Ground truth agents | Ni | Ni |
| Predicted solvents |  ✅ (yellow) |  ❌ |
| Predicted agents | Pd ❌ | Pd ❌ |

**Reaction type**: Corey Seebach reaction
**Comment**: Ethanol is categorized either as solvent or reactant - both roles makes sense chemically. Within the prediction, trust labelling predicted ethyl acetate which is uncommon for this transformtation. Using the reaction string dataset, THF was predicted which is correct, however, the initiall data also contained EtOH. Pd has been predicted in both cases as agent which is incorrect.

✔ Correct prediction     ✖ Incorrect prediction     ✔ Partially correct prediction

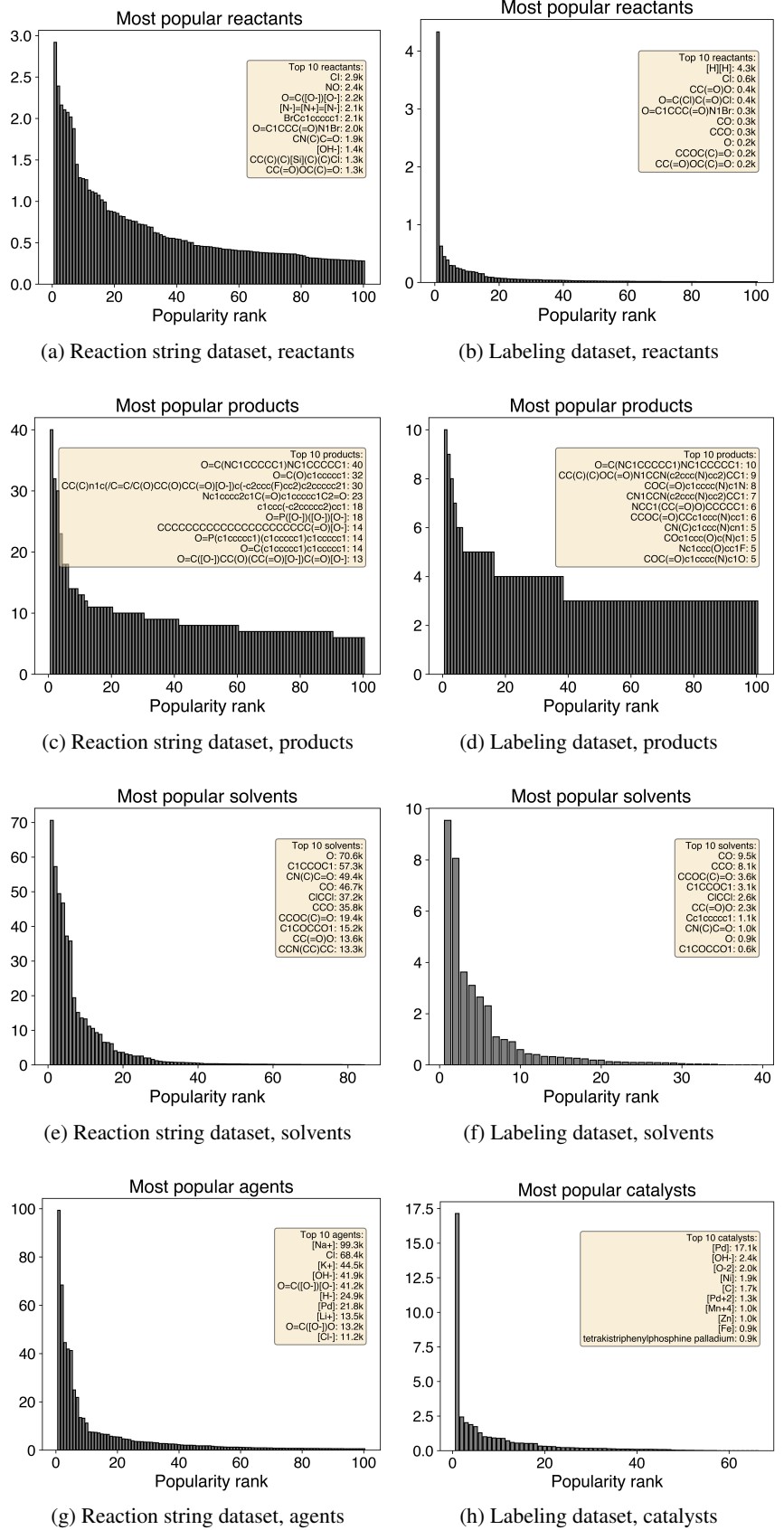

Figure 7: Frequency of occurrence of the most popular molecules. NULL has been removed, reagents and catalysts have been merged.

## E   Datasheet for ORDerly dataset

### E.1   Motivation

Q1: **For what purpose was the dataset created?** Was there a specific task in mind? Was there a specific gap that needed to be filled? Please provide a description.

- The datasets were created to facilitate building machine learning models for prediction of reaction products, retrosynthesis, and reaction conditions in chemical synthesis, particularly in the context of the pharmaceutical industry. There was a need of a clean, high-quality reaction condition benchmark dataset, in addition to a need for an open-source repository for cleaning reactions, and an investigation of how decisions made during cleaning impact the usefulness of the model that is trained on the datasets. ORDerly solves all three of these issues. The code for ORDerly, and the raw data used to generate the ORDerly benchmark datasets, are both open-source, making the benchmark generation accessible and reproducible.

Q2: **Who created the dataset (e.g., which team, research group) and on behalf of which entity (e.g., company, institution, organization)?**

- ORDerly was built by researchers from the group of Anon at Institution.

Q3: **Who funded the creation of the dataset?** If there is an associated grant, please provide the name of the grantor and the grant name and number.

- This work is co-funded by UCB Pharma and Engineering and Physical Sciences Research Council via project EP/S024220/1 EPSRC Centre for Doctoral Training in Automated Chemical Synthesis Enabled by Digital Molecular Technologies. This project was co-funded by European Regional Development Fund via the project "Innovation Centre in Digital Molecular Technologies".

Q4: **Any other comments?**

- No.

### E.2   Composition

Q5: **What do the instances that comprise the dataset represent (e.g., documents, photos, people, countries)?** *Are there multiple types of instances (e.g., movies, users, and ratings; people and interactions between them; nodes and edges)? Please provide a description.*

- Eight datasets were presented in this work. Each dataset was saved in Apache Parquet format, and has the following column groups:
    * Reaction SMILES string (string), is_mapped (bool)
    * Reactants & products (SMILES strings)
    * Solvents and agents (rxn string data), or solvents, catalysts, and reagents (labeling data) (SMILES strings)
    * Temperature, reaction time, yield (floats)
    * Procedure details (string)
    * Grant date (datetime), date of experiment (datetime), file name (string)

Q6: **How many instances are there in total (of each type, if appropriate)?**

- The number of reactions in each dataset is outlined in detail in Table 1.

Q7: **Does the dataset contain all possible instances or is it a sample (not necessarily random) of instances from a larger set?** *If the dataset is a sample, then what is the larger set? Is the sample representative of the larger set (e.g., geographic coverage)? If so, please describe how this representativeness was validated/verified. If it is not representative of the larger set, please describe why not (e.g., to cover a more diverse range of instances, because instances were withheld or unavailable).*

- All the data in ORD was used to generate the datasets presented in this paper. Datasets A-F were built from the subset of ORD belonging to USPTO (1.7m reactions in total), while datasets G-H were built on the subset of data from ORD that do not belong to USPTO (91k reactions in total, as of August 2023).

Q8: **What data does each instance consist of?** *"Raw" data (e.g., unprocessed text or images) or features? In either case, please provide a description.*

- Chemical reaction data stored in ORD is structured like a json/dictionary, with strings and floats as the values. The values that are relevant to ORDerly were discussed in response to Q5. A full description of the data stored in ORD is available elsewhere [20].

Q9: **Is there a label or target associated with each instance?** *If so, please provide a description.*

- There is a label associated with molecules in ORD, and in this work we show the pitfalls of relying on this label, and present ORDerly to more robustly assign labels. The targets are the reaction conditions (solvents, agents, catalysts, reagents).

Q10: **Is any information missing from individual instances?** *If so, please provide a description, explaining why this information is missing (e.g., because it was unavailable). This does not include intentionally removed information, but might include, e.g., redacted text.*

- Many reactions were missing temperature, reaction time, and yield data; this is likely due to this information not being recorded by the experimentalist, or not extracted when the information was scraped from a patent/paper.

Q11: **Are relationships between individual instances made explicit (e.g., users' movie ratings, social network links)?** *If so, please describe how these relationships are made explicit.*

- Each row contains information for a single step chemical reaction. The only explicit link between reactions is the year they were performed or the year that the corresponding patent was granted. The year a chemical reaction was performed may imply some degree of chemical information, since chemical reactions of a certain type obviously could not have been performed before they were invented. Furthermore, "hype" around a particular type of reaction may influence how often certain reaction classes are used through time. For these reasons, a time-based split can be viewed as a (somewhat poor) proxy for a reaction class split. There is a column in the dataset containing the year that the grant was awarded, and another column for time of experiment.

Q12: **Are there recommended data splits (e.g., training, development/validation, testing)?** If so, please provide a description of these splits, explaining the rationale behind them.

- We recommend using a random split of the ORDerly benchmarks, and provide pre-split data to ensure that ML researchers using the benchmark use the same train/test split. There are three data splits that would make sense on a chemical reactions dataset: a random split, a time split, and the reaction class split. A reaction class split would require models to generalise to unseen reaction classes (as opposed to unseen reactions of the same class), making the prediction task much more difficult. As explained above (Q16), using a time split would effectively just serve as a proxy for a reaction class split, and is therefore not desirable. There are a number of reasons for the random split being preferred over the reaction class split: 1) A reaction class split would need to either use an ML clustering algorithm (which usually work quite well, but cannot be viewed as a ground-truth split), or using proprietary software based on manually curated chemistry rules (which would mean that the full pipeline is no longer fully open source and reproducible). 2) The reaction prediction task is already difficult enough with a random split (e.g. considering our top-3 accuracy of sub 50%, and models trained on a random split are still able to provide value even if they can only make predictions on reaction classes that they have seen before - the reaction classes represented in the dataset will likely be the most popular reaction classes, and therefore also those most likely to be queried by the end user.

Q13: **Are there any errors, sources of noise, or redundancies in the dataset?** *If so, please provide a description.*

- The `ORDerly-condition`, `ORDerly-forward`, and `ORDerly-retro` datasets are generated from the USPTO dataset, which is a dataset made from chemical reactions from US Patents. When a molecule is patented, it is also a requirement to publish the synthesis pathway to produce the molecule, and it is from these synthesis pathways that reactions are extracted. To avoid giving away proprietary information there is an incentive to use already published "industry standard" reaction conditions in the patent application; furthermore, the "first to file" nature of the US patent system means there is an incentive to apply for patents as soon as possible. These two factors may bias the reactions in the USPTO dataset towards being unoptimized, low-yielding reactions that can also be found elsewhere. In fact, we observed that $\approx 40\%$ of reactions were dropped because they were duplicates (see Table 1), indicating that many reactions are executed at "standard conditions" for a particular class of reaction instead of being optimized for the specific reactants.
- Reproducibility is known to be difficult in chemistry[44], which implies a base-level of noise in the dataset.

Q14: **Is the dataset self-contained, or does it link to or otherwise rely on external resources (e.g., websites, tweets, other datasets)?** *If it links to or relies on external resources, a) are there guarantees that they will exist, and remain constant, over time; b) are there official archival versions of the complete dataset (i.e., including the external resources as they existed at the time the dataset was created); c) are there any restrictions (e.g., licenses, fees) associated with any of the external resources that might apply to a future user? Please provide descriptions of all external resources and any restrictions associated with them, as well as links or other access points, as appropriate.*

- The ORDerly datasets are self-contained. To be able to reproduce cleaning of ORD data, the ORD data will naturally need to continue to exist. ORD was built to be an open-source tool, so there should not be any restrictions on its use in the future.

Q15: **Does the dataset contain data that might be considered confidential (e.g., data that is protected by legal privilege or by doctor–patient confidentiality, data that includes the content of individuals' non-public communications)?** *If so, please provide a description.*

- No.

Q16: **Does the dataset contain data that, if viewed directly, might be offensive, insulting, threatening, or might otherwise cause anxiety?** *If so, please describe why.*

- No.

Q17: **Does the dataset relate to people?** *If not, you may skip the remaining questions in this section.*

- No.

Q18: **Does the dataset identify any subpopulations (e.g., by age, gender)?**

- No.

Q19: **Is it possible to identify individuals (i.e., one or more natural persons), either directly or indirectly (i.e., in combination with other data) from the dataset?** *If so, please describe how.*

- No.

Q20: **Does the dataset contain data that might be considered sensitive in any way (e.g., data that reveals racial or ethnic origins, sexual orientations, religious beliefs, political opinions or union memberships, or locations; financial or health data; biometric or genetic data; forms of government identification, such as social security numbers; criminal history)?** *If so, please provide a description.*

- No.

Q21: **Any other comments?**

   - No.

## E.3 Collection process

Q22: **How was the data associated with each instance acquired?** Was the data directly observable (e.g., raw text, movie ratings), reported by subjects (e.g., survey responses), or indirectly inferred/derived from other data (e.g., part-of-speech tags, model-based guesses for age or language)? If data was reported by subjects or indirectly inferred/derived from other data, was the data validated/verified? If so, please describe how.

   - The raw data of each instance (reaction) was extracted from United States Patents to form the "USPTO dataset" [21]. The USPTO dataset was parsed into ORD format [20], where we extracted it from. ORD does contain additional data, beyond the USPTO dataset. Other reactions in ORD are contributed by chemists in academia and industry.

Q23: **What mechanisms or procedures were used to collect the data (e.g., hardware apparatus or sensor, manual human curation, software program, software API)?** How were these mechanisms or procedures validated?

   - Data in the ORD database is readily downloadable through the GitHub repository.

Q24: **If the dataset is a sample from a larger set, what was the sampling strategy (e.g., deterministic, probabilistic with specific sampling probabilities)?**

   - See Q7.

Q25: **Who was involved in the data collection process (e.g., students, crowdworkers, contractors) and how were they compensated (e.g., how much were crowdworkers paid)?**

   - N/A.

Q26: **Over what timeframe was the data collected? Does this timeframe match the creation timeframe of the data associated with the instances (e.g., recent crawl of old news articles)?** *If not, please describe the timeframe in which the data associated with the instances was created.*

   - The reactions in the USPTO dataset are from patents which were published between 1976 and September 2016. The USPTO dataset was parsed into ORD in 2020. Additional reactions not from patents have since been added to ORD. ORDerly was built in 2023.

Q27: **Were any ethical review processes conducted (e.g., by an institutional review board)?** *If so, please provide a description of these review processes, including the outcomes, as well as a link or other access point to any supporting documentation.*

   - No.

Q28: **Does the dataset relate to people?** *If not, you may skip the remaining questions in this section.*

   - No.

Q29: **Did you collect the data from the individuals in question directly, or obtain it via third parties or other sources (e.g., websites)?**

   - N/A.

Q30: **Were the individuals in question notified about the data collection?** *If so, please describe (or show with screenshots or other information) how notice was provided, and provide a link or other access point to, or otherwise reproduce, the exact language of the notification itself.*

   - N/A.

Q31: **Did the individuals in question consent to the collection and use of their data?** *If so, please describe (or show with screenshots or other information) how consent was requested and provided, and provide a link or other access point to, or otherwise reproduce, the exact language to which the individuals consented.*

– N/A.

Q32: **If consent was obtained, were the consenting individuals provided with a mechanism to revoke their consent in the future or for certain uses?** *If so, please provide a description, as well as a link or other access point to the mechanism (if appropriate).*

– N/A.

Q33: **Has an analysis of the potential impact of the dataset and its use on data subjects (e.g., a data protection impact analysis) been conducted?** *If so, please provide a description of this analysis, including the outcomes, as well as a link or other access point to any supporting documentation.*

– N/A.

Q34: **Any other comments?**

– No.

## E.4   Preprocessing, cleaning, and/or labeling

Q35: **Was any preprocessing/cleaning/labeling of the data done (e.g., discretization or bucketing, tokenization, part-of-speech tagging, SIFT feature extraction, removal of instances, processing of missing values)?** If so, please provide a description. If not, you may skip the remainder of the questions in this section.

– Yes, this is described in detail in section 4 and A.

Q36: **Was the "raw" data saved in addition to the preprocessed/cleaned/labeled data (e.g., to support unanticipated future uses)?** If so, please provide a link or other access point to the "raw" data.

– The raw structured data is stored in the ORD GitHub repository.

Q37: **Is the software used to preprocess/clean/label the instances available?** If so, please provide a link or other access point.

– This paper is for the software used to preprocess, clean, and label the instances.

Q38: **Any other comments?**

– No.

## E.5   Uses

Q39: **Has the dataset been used for any tasks already?** *If so, please provide a description.*

– Yes, in section 5 we train a previously published neural network model for reaction condition prediction and a previously published transformer for forward prediction and retrosynthesis.

Q40: **Is there a repository that links to any or all papers or systems that use the dataset?** If so, please provide a link or other access point.

– No.

Q41: **What (other) tasks could the dataset be used for?**

– As described in section 2, other key problems in chemical synthesis include reaction outcome prediction, retrosynthesis, and reaction condition prediction. An important task which was not described is reaction yield prediction. Successful reaction yield models are predominantly trained on high-throughput experimentation (HTE) datasets [15], and is known to be difficult (if not impossible) with patent data (e.g. USPTO)

[13, 14]. As long as ORD primarily consists of USPTO data, ORDerly will probably not be very useful for yield prediction, but it could be in the future.

Q42: **Is there anything about the composition of the dataset or the way it was collected and preprocessed/cleaned/labeled that might impact future uses?** *For example, is there anything that a future user might need to know to avoid uses that could result in unfair treatment of individuals or groups (e.g., stereotyping, quality of service issues) or other undesirable harms (e.g., financial harms, legal risks) If so, please provide a description. Is there anything a future user could do to mitigate these undesirable harms?*

– Yes, ORDerly relies on the ORD schema, and changes to the ORD schema or ORD database may require updates to ORDerly. ORD may change in the future, as the it becomes more clear how the community wishes to use ORD (e.g. which classes of information are stored).

Q43: **Are there tasks for which the dataset should not be used?** *If so, please provide a description.*

– The ORDerly datasets were generated to make it easier to train models that can predict how to make small molecules. The intended usage is to predict synthesis pathways for therapeutics, however, within this category of small molecules is also energetic materials, such as explosives.

Q44: **Any other comments?**

– No.

### E.6    Distribution

Q45: **Will the dataset be distributed to third parties outside of the entity (e.g., company, institution, organization) on behalf of which the dataset was created?** If so, please provide a description.

– Yes, the datasets will be open-source.

Q46: **How will the dataset be distributed (e.g., tarball on website, API, GitHub)?** Does the dataset have a digital object identifier (DOI)?

– The data is available through FigShare. (https://doi.org/10.6084/m9.figshare.23298467)
– It can also reliably be recreated using the instructions in the ORDerly GitHub repository (https://github.com/sustainable-processes/ORDerly).

Q47: **When will the dataset be distributed?**

– It is already publicly available.

Q48: **Will the dataset be distributed under a copyright or other intellectual property (IP) license, and/or under applicable terms of use (ToU)?** *If so, please describe this license and/or ToU, and provide a link or other access point to, or otherwise reproduce, any relevant licensing terms or ToU, as well as any fees associated with these restrictions.*

– CC-BY-4.0

Q49: **Have any third parties imposed IP-based or other restrictions on the data associated with the instances?** *If so, please describe these restrictions, and provide a link or other access point to, or otherwise reproduce, any relevant licensing terms, as well as any fees associated with these restrictions.*

– No.

Q50: **Do any export controls or other regulatory restrictions apply to the dataset or to individual instances?** *If so, please describe these restrictions, and provide a link or other access point to, or otherwise reproduce, any supporting documentation.*

– No,

Q51: **Any other comments?**

– No.

## E.7 Maintenance

Q52: **Who will be supporting/hosting/maintaining the dataset?**

– The dataset is hosted on FigShare, the code to generate the dataset is hosted on GitHub.
– The group of Anon will be maintaining ORDerly.

Q53: **How can the owner/curator/manager of the dataset be contacted (e.g., email address)?**

– Anon.

Q54: **Is there an erratum?** If so, please provide a link or other access point.

– N/A.

Q55: **Will the dataset be updated (e.g., to correct labeling errors, add new instances, delete instances)?** *If so, please describe how often, by whom, and how updates will be communicated to users (e.g., mailing list, GitHub)?*

– ORDerly will be maintained by the group of Anon, updates will be tracked through GitHub. ORDerly is built to be extensible, such that as the ORD dataset grows, users can run ORDerly to create new, larger, datasets. The ORDerly benchmark datasets are unlikely to change (to ensure model accuracy is comparable).

Q56: **If the dataset relates to people, are there applicable limits on the retention of the data associated with the instances (e.g., were individuals in question told that their data would be retained for a fixed period of time and then deleted)?** *If so, please describe these limits and explain how they will be enforced.*

– N/A.

Q57: **Will older versions of the dataset continue to be supported/hosted/maintained?** If so, please describe how. If not, please describe how its obsolescence will be communicated to users.

– The datasets are small enough to easily be versioned and hosted on FigShare (350k-1m reactions, 200MB-500MB).

Q58: **If others want to extend/augment/build on/contribute to the dataset, is there a mechanism for them to do so?** *If so, please provide a description. Will these contributions be validated/verified? If so, please describe how. If not, why not? Is there a process for communicating/distributing these contributions to other users? If so, please provide a description*

– All contributions to ORDerly will be managed through the ORDerly GitHub repository. Pull requests into main will need to be verified by a member of Anon's group.

Q59: **Any other comments?**

– No.

