# OpenReview forum: "ORDerly: Datasets and benchmarks for chemical reaction data"
_NeurIPS.cc/2023/Workshop/AI4Science — NeurIPS2023-AI4Science Poster_

### Official Review · Reviewer_R5EB · 2023-10-23
**Review of ORDerly: Datasets and benchmarks for chemical reaction data**

**Rating:** 7
**Confidence:** 4

**Review:**

This manuscript proposed a benchmarking effort for chemical reaction condition prediction, called ORDerly. The core of ORDerly is data extraction and data cleaning from the existing Open reaction Database (ORD). By experimenting with different ways to label molecules’ role in reaction and treating rare agents, authors obtained 4 sets of data from ORD. They then tested Gao et al.’s model on each dataset, and the results suggest that the original labeling within ORD could be enhanced, and the removal of infrequently occurring agents can notably improve the model's performance. The task is useful and important, and the proposed data and benchmark seems reasonable.

---

### Meta-Review · Area_Chair_5qnN · 2023-10-26

**Recommendation:** Accept (Poster)
**Confidence:** 4

**Metareview:**

Good paper. Accept